# Explaining the Hardening Structures of Helium Spectrum and Boron to Carbon Ratio through Different Propagation Models

Qing Luo [1], Jie Feng [2,*] and Pak-Hin Thomas Tam [1,*]

1   School of Physics and Astronomy, Sun Yat-sen University, Zhuhai 519082, China
2   School of Science, Shenzhen Campus of Sun Yat-sen University, Shenzhen 518107, China
*   Correspondence: fengj77@sysu.edu.cn (J.F.); tanbxuan@sysu.edu.cn (P.-H.T.T.)

**Abstract:** Recently, a series of high-precision measurements by various experiments show that cosmic ray nuclei spectra begin to harden at ∼200 GV and the boron-to-carbon (B/C) ratio has a similar trend around the same energy. These anomalous structures possibly result from the journey of cosmic rays (CRs) from their sources to our solar system, which has important implications for our understanding of the origin and propagation of Galactic cosmic rays (GCRs). In this work, we investigate several propagation models and attempt to explain these anomalous observations. We have verified that an extension of the traditional propagation model taking into account spatially dependent propagation and secondary particle acceleration provides a more accurate description of the latest B/C ratio and the Helium flux data measured by DAMPE, CALET, and AMS-02.

**Keywords:** galactic cosmic rays; spectra; B/C ratio; propagation parameters

## 1. Introduction

Cosmic rays (CRs) are high-energy charged particles from the universe, which are usually classified into two categories, galactic cosmic rays (GCRs) and extragalactic cosmic rays (EGCRs), according to their energy [1]. Supernova remnants (SNRs) are generally believed to be the primary source of CRs in the Milky Way [2–4]. After being accelerated near the source region, the primary CRs are injected into interstellar space, where they undergo a series of processes, such as diffusion, convection, re-acceleration, energy loss, fragmentation, decay, and so on. Concurrently, the secondary particles are generated by the interaction between the propagating primary CRs and the interstellar medium (ISM). Then, these primary and secondary particles reach the solar system, are further deflected by the solar magnetic field, and are finally detected by space satellites.

According to the usual Fermi acceleration model, the energy spectrum of cosmic ray particles is expected to be a single power law spectrum, which begins to soften after diffusion in the interstellar turbulent magnetic field. However, such a simple image has recently been challenged by some high-precision measurements which show more complex observations. Protons are the most abundant charged particles of CRs. Explaining the variation in their energy spectra is very important for understanding the origin, acceleration, and propagation of CRs. First, the Advanced Thin Ionization Calorimeter (ATIC-2), Cosmic Ray Energetics and Mass (CREAM), and Payload for Antimatter Matter Exploration and Light-nuclei Astrophysics (PAMELA) experiments [5–9] revealed the fine structure of spectral hardening of the proton at ∼200 GV. The Alpha Magnetic Spectrometer (AMS-02) experiment has, for the first time, provided a detailed analysis of the rigidity dependence of the spectral features in CR nuclei spectra and confirmed that fine structure with unprecedented precision [10–12]. The Calorimetric Electron Telescope (CALET) has not only confirmed the results but has for the first time extended the measurement of protons in CRs from a few tens of GeV to 10 TeV consistently with one instrument [13]. More interestingly, the CREAM, NUCLEON, and Dark Matter Particle Explorer (DAMPE) experiments observed a

spectral break at about 14 TeV [8,14–17]. In addition to the new hardening and softening characteristics found in the proton spectrum, similar structural changes were also found in the helium spectrum. The NUCLEON, PAMELA, and AMS-02 experiments found that the proton spectrum has different energy indices than the helium one, and that the spectrum for helium hardens at the same energy as the one for protons [9,10,14]. The softening of helium nuclear spectrum at 34 TeV was found for the first time by DAMPE experiment [17]. The similar results measured by these experiments all indicate that there are new characteristics of the cosmic ray energy spectra in the lower than the knee region, and these characteristics can provide new clues for the study of the origin of GCRs.

In addition, accurate measurement of the ratio between secondary particles and primary particles are also an important way to determine GCR propagation and the turbulence characteristics of the ISM [18–20]. The boron–carbon ratio (B/C) is the best measured and most widely studied among the various secondary to primary nuclear ratios. Many previous experiments [11,21–32] have given the measurements of B/C within TeV energy, especially the high accuracy of AMS-02, and these experimental results have been widely used in the study of GCRs propagation models [11,21,27,30,31]. Recently, DAMPE extended the high-precision measurement of B/C to 5 TeV/n and found that the measured B/C showed significant hardening at kinetic energy (∼100 GeV/n) [33]. Previous measurements by AMS-02 and other experiments also showed that the energy spectra of primary nuclei also hardened at similar energies significantly [6,7,9,10,16,17,34]. These observations indicate that all CR species have experienced some common mechanisms, during their acceleration at sources or propagation in the Galaxy.

Several models have been proposed to explain these anomalous observations, such as the nearby source [35–38], the combined effects from different group sources [39], and spatially dependent propagation (SDP) of CRs [40–43], et al. In this work, we mainly investigate three propagation models, including the traditional propagation (TP) model, SDP model and the neighboring source accelerated secondary particle (Fresh) model, to try to explain the Helium spectrum and the latest B/C ratio observed by DAMPE. The explanation of these three models will be described in detail later in the text.

The paper is organized as the following. Section 2 introduces the CR propagation models that are commonly used in the community. Section 3 shows the model calculations compared with experimental data. Finally, we will draw our conclusions in Section 4.

## 2. Models

CRs are accelerated near the source and then injected into interstellar space, which we call primary cosmic rays. After entering interstellar space, cosmic rays will be frequently scattered by irregular magnetic fields, and this kind of motion is an approximately random diffusion process. Then, cosmic rays will also interact with ISM during their propagation, losing some energy through synchrotron radiation, inverse Compton scattering, Bremsstrahlung radiation, and other processes. In addition, cosmic rays may also collide with gaseous materials in the process of fragmentation, which will create new secondary particles. Finally, these particles will be further affected by the solar wind [44] and its magnetic field before reaching Earth to be observed.

We can use a propagation equation to describe the processes of diffusion, convection, energy loss, diffusion reacceleration, fragmentation, and annihilation of cosmic rays as they propagate in the interstellar medium [45]. The complete propagation equation is as follows:

$$
\begin{aligned}
\frac{\partial \psi(\vec{r},p,t)}{\partial t} &= q(\vec{r},p,t) + \vec{\nabla} \cdot \left( D_{xx} \vec{\nabla} \psi - \vec{v}_c \psi \right) \\
&+ \frac{\partial}{\partial p} p^2 D_{pp} \frac{\partial}{\partial p} \frac{1}{p^2} \psi - \frac{\partial}{\partial p} \left[ \dot{p}\psi - \frac{p}{3} \left( \vec{\nabla} \cdot \vec{v}_c \psi \right) \right] , \\
&- \frac{\psi}{\tau_f} - \frac{\psi}{\tau_r}
\end{aligned}
\tag{1}
$$

where $\psi(\vec{r}, p, t)$ is the density of CR particles per unit momentum $p$ at position $\vec{r}$, $q(\vec{r}, p, t)$ the acceleration sources, $D_{xx}$ and $D_{pp}$ the diffusion coefficients in the coordinate and the

momentum space, respectively, $\vec{v}_c$ the convection velocity, $\dot{p} \equiv \mathrm{d}p/\mathrm{d}t$ the momentum loss rate, $\tau_f$ and $\tau_r$ the characteristic time scales for fragmentation and radioactive decay, respectively.

For the lithium, beryllium and boron, generated during cosmic ray propagation, the so-called straight-ahead approximation is generally used [46,47], where kinetic energy per nucleon is conserved during the interactions. The production rate is thus

$$Q_j = \sum_{i=C,N,O} (n_H \sigma_{i+H \to j} + n_{He} \sigma_{i+He \to j}) v \psi_i \quad , \tag{2}$$

where $n_{H/He}$ is the number density of hydrogen/helium in the ISM and $\sigma_{i+H/He \to j}$ is the total cross-section of the corresponding hadronic interaction.

### 2.1. Traditional Propagation Model

The region of random motion of GCRs is called the diffusion halo, which is usually approximated to be a cylinder with a radial radius of $R = 20$ kpc and vertical half thickness of $z_h = 5$ kpc. The Galactic disk is located in the middle of the diffusion halo with a thickness of ~200 pc. The sun is located in the Galactic disk and about 8.5 kpc from the Galactic Center (GC). In the study of cosmic ray propagation, it is generally believed that cosmic rays spread uniformly and isotropic, and the diffusion coefficient only depends on the rigidity, expressed as

$$D_{xx}(\mathcal{R}) = D_0 \beta^\eta \left( \frac{\mathcal{R}}{\mathcal{R}_0} \right)^\delta \quad , \tag{3}$$

where $D_0$ is the normalized coefficient, $\beta^\eta$ describes the low-energy diffusion [48], $\beta$ is the velocity of the particle in the unit of light speed, $\eta$ is a constant, $\mathcal{R} \equiv p/Z$ the rigidity, $\mathcal{R}_0$ the reference rigidity, $\delta$ the power-law index, related to the interstellar turbulence. In addition, random magnetohydrodynamic waves can also generate random acceleration of cosmic rays. The detailed process is related to the particle and wave collision rate and the plasma turbulence velocity, Alfvén velocity $v_A$. According to the quasi-linear theory [49], the diffusion coefficient of momentum space is expressed as

$$D_{pp} = \frac{4 p^2 v_A^2}{3 \delta (4 - \delta^2)(4 - \delta) D_{xx}} \quad , \tag{4}$$

The convective effects of stellar winds, supernova explosions, and cosmic rays themselves on cosmic rays are not considered in this work.

The distribution of the SNRs is approximated as axisymmetric, which is usually parameterized as

$$f(r,z) = \left( \frac{r}{R_\odot} \right)^a \exp \left[ -\frac{b(r - R_\odot)}{R_\odot} \right] \exp \left( -\frac{|z|}{z_s} \right) \quad , \tag{5}$$

where $R_\odot = 8.5$ kpc represents the distance from the solar system to the Galactic center. The parameters $a$ and $b$ are set to be 1.69 and 3.33 [50], respectively. The density distribution of the SNRs decreases exponentially along the vertical height from the Galactic plane, with $z_s = 200$ pc. In this work, we assume the injection spectrum as an exponentially cut-off broken power-law function of particle rigidity $\mathcal{R}$ [40]

$$q(\mathcal{R}) = q_0 \begin{cases} \left( \dfrac{\mathcal{R}}{\mathcal{R}_{\mathrm{br}}} \right)^{\nu_1}, & \mathcal{R} \leq \mathcal{R}_{\mathrm{br}} \\[2ex] \left( \dfrac{\mathcal{R}}{\mathcal{R}_{\mathrm{br}}} \right)^{\nu_2} \exp \left[ -\dfrac{\mathcal{R}}{\mathcal{R}_{\mathrm{c}}} \right], & \mathcal{R} > \mathcal{R}_{\mathrm{br}} \end{cases} \quad , \tag{6}$$

where $q_0$ is the normalization factor, $\nu_{1,2}$ the spectral indices, $\mathcal{R}_{br}$ is the break rigidity, and $\mathcal{R}_c$ is the cutoff rigidity. To fit the low energy measurements, we use the force field approximation to describe the solar modulation effect of GCRs [51].

## 2.2. Spatially Dependent Propagation

In the traditional propagation model, the diffusion coefficient is uniform. However, recent observations of high-energy extended gamma-ray halos around some pulsars by the High-Altitude Water Cherenkov Observatory (HAWC) and the Large High-Altitude Air Shower Observatory (LHAASO) suggest that the diffusion coefficient for charged particles in the ISM surrounding some pulsars is smaller [52,53]. Combined with the diffusion coefficient inferred from the secondary to primary ratio directly measured by GCR, the propagation of GCR is likely to be uneven—slow in the galactic disk (or near the source) and faster in the halo [54,55].

In explaining the spectral hardening of protons and helium nuclei at 200 GV, Tomassetti pioneered a two-halo model (THM) in 2012 [56], introducing the concepts of inner halo (IH) and outer halo (OH). The spatially dependent propagation (SDP) model further calibrates the diffusion coefficient according to the real distribution of the source. In the SDP model, the diffusion halo is divided into two regions according to the diffusion characteristics. The inner halo region (IH) is closer to the galactic disk, with more sources distributed. Affected by the supernova explosion, the turbulence is more intense, and the diffusion process is relatively slow. So the diffusion coefficient is more dependent on the distribution of the source. In the outer halo region (OH), the diffusion process is faster due to the relatively mild turbulence and fewer source distributions. Therefore, the diffusion coefficient of the OH region is only related to the rigidity, just like the traditional model. The diffusion coefficient D for the whole region is parameterized [40,43,48],

$$D_{xx}(r,z,\mathcal{R}) = D_0 F(r,z)\beta^\eta \left(\frac{\mathcal{R}}{\mathcal{R}_0'}\right)^{\delta(r,z)} , \qquad (7)$$

$$\delta(r,z) = \delta_0 F(r,z) , \qquad (8)$$

$$F(r,z) = \begin{cases} g(r,z) + [1-g(r,z)](\frac{z}{\xi z_h})^n, & |z| \le \xi z_h \\ 1, & |z| > \xi z_h \end{cases} , \qquad (9)$$

Here, $\delta(r,z)$ (or $\delta_0$) reflects the property of irregular turbulence int IH (or OH), $g(r,z) = N_m/[1+f(r,z)]$, in which $f(r,z)$ is the source distribution in Equation (5). $\xi z_h$ and $(1-\xi)z_h$ represent the size of the IH and OH area, respectively, and the factor $\frac{z}{\xi z_h}^n$ is used to describe the smooth connection between two halos and can be fixed as is the case in [41,57]. $R_0'$ is a constant that we usually set $R_0' = 4$ GV, and $N_m$ is a free parameter used in source distribution. For more questions about this model, please refer to [40].

## 2.3. SDP+Fresh

In conventional propagation scenarios, secondary particles are produced by the inelastic interaction of primary GCRs with ISM as they propagate through the galaxy [58]. However, the newly accelerated CRs can also interact with the surrounding gas to produce secondary particles before escaping the source region and entering the diffusion halo. Previous work has generally overlooked this contribution, but, in fact, this contribution may not be small [59] and is expected to become increasingly important at high energies because the CR spectrum around the source is harder than the CR spectrum spreading through the Milky Way. Therefore, based on the SDP model, we reconsidered this contribution and called it the "fresh" component.

The yield of secondary B produced by the newly accelerated CRs interacting with the surrounding gas near the source can be calculated as

$$Q_j = \sum_{i=C,N,O}(n_H \sigma_{i+H \to j} + n_{He} \sigma_{i+He \to j})v Q_i(E)t_{col} \quad, \tag{10}$$

where $n_{H/He}$ is the number density of hydrogen/helium, $Q_i(E)$ is the function of the injection source, and $t_{col}$ is the duration time of collisions between the primary CRs and molecular gas.

## 3. Results

Helium, carbon, and oxygen are among the primary nuclei in cosmic rays. The measurement results of AMS-02 show that the fluxes of these three elements deviate from a single power law, and the three fluxes have identical rigidity dependence above 60 GV [10]. One of them can be used as the observable of primary CRs. The flux of Helium has the lowest uncertainties, so it is chosen to better constrain the parameter models tested here. In this work, we discuss and compare these three models in terms of the secondary-to-primary ratio (represented by B/C ratio) and energy spectrum. Although this work is presently limited to the analysis of the Helium flux, the extension of the analysis to other nuclei fluxes, such as carbon, oxygen, etc., will be the objective of a future publication.

In this work, we employ the publicly available numerical code DRAGON, which was designed to solve the propagation of CRs. The introduction of DRAGON and the calculation of the energy spectrum are described in detail in Appendix A. First, we use the data of B/C to constrain the relevant parameters of CR transport for different models, namely, $D_0$, $\delta_0$, and $N_m$. In addition, by adopting one set of diffusion coefficients, we make the calculation of the Helium spectrum, which introduces five parameters describing the background injection: the parameter $Q_0$ for the injection power, the parameters $\nu_1$ and $\nu_2$ for the helium injection indexes, the parameters $R_{br}$ and $R_c$ for the break rigidity and cut-off rigidity, respectively; and three parameters describing the local sources injection: the parameter $q_0$ for the injection power, the parameter $\gamma$ for the local source injection index, the parameter $R_c$ for the cut-off rigidity. Finally, we compare the model calculations with the observed results. The detailed propagation and injection parameters are listed in Tables 1 and 2. In addition, when fitting the B/C ratio, the corresponding parameters of the approximate force field used by the three models are 0.6 GeV, 0.6 GeV, and 0.7 GeV, respectively. The parameters of the force field approximately used in fitting the helium spectrum are 0.65 GeV, 0.6 GeV, and 0.72 GeV. This work provides a first comparison of different propagation models and does not discuss at all the goodness of the fit, the model parameter uncertainties, and how statistical and systematic uncertainties (which could introduce correlations between data points) of the B/C and He flux data have been used in the fitting procedure. We are working on the Bayesian analyses of the fluxes with different models, which will be discussed in future publications.

**Table 1.** The propagation parameters of three different models discussed in Section 2.

| Model | $D_0$ $[\text{cm}^2 \cdot \text{s}^{-1}]$ | $\delta_0$ | $N_m$ | $z_h$ [kpc] | $\xi$ | $v_A$ $[\text{km} \cdot \text{s}^{-1}]$ | n | $\eta$ | $\mathcal{R}_0$ |
|---|---|---|---|---|---|---|---|---|---|
| TP | $3.82 \times 10^{28}$ | 0.49 | | 5 | | 6 | 4 | 0.05 | 4 |
| SDP | $5.24 \times 10^{28}$ | 0.63 | 0.33 | 5 | 0.1 | 6 | 4 | 0.05 | 4 |
| SDP+Fresh | $4.77 \times 10^{28}$ | 0.64 | 0.58 | 5 | 0.1 | 6 | 4 | 0.05 | 4 |

**Table 2.** The Background and Local SNRs-Injected Parameters for helium.

| Model | $Q_0$ $[\text{m}^{-2}\text{sr}^{-1}\text{s}^{-1}\text{GeV}^{-1}]$ | $\nu_1$ | $\nu_2$ | $\mathcal{R}_{br}$ [GV] | $\mathcal{R}_c$ [TV] | $q_0$ $[\text{GeV}^{-1}]$ | $\gamma$ | $\mathcal{R}_c$ [TV] |
|---|---|---|---|---|---|---|---|---|
| | | | Background | | | | Local | |
| TP | 4.75 | 2.3 | 2.42 | 6.8 | 150 | | | |
| SDP | 4.62 | 2 | 2.54 | 6 | 200 | | | |
| SDP+Fresh | 4.02 | 2 | 2.39 | 6 | 4 | 7000 | $2.16 \times 10^{52}$ | 28 |

### 3.1. B/C Ratio

The black lines in Figure 1 show the B/C ratios calculated by three different propagation models. The dotted-solid line represents the expectation for the TP model, the dashed line is the result of the SDP model, and the solid line displays the SDP+Fresh model. The red dots, green squares, and blue triangles in Figure 1 represent the B/C ratios' measurements obtained by AMS-02, DAMPE, and CALET experiments, respectively. The results of AMS-02 and CALET are more similar, while the results of DAMPE are higher. It should be noted that the purpose of our work is to explain the hardening of the B/C ratio, and the low energy data can be appropriately ignored, so the expected results of the three models are drawn from the place of 8 GeV/n. It can be seen that the TP model can explain the B/C ratio below ∼100 GeV/n well, but it is not sufficient to reproduce the hardening above ∼100 GeV/n hinted by DAMPE, CALET and AMS-02 data. However, the SDP and SDP+fresh models can both reproduce the B/C ratio hardening, especially for the DAMPE measurement, within the errors at high energies.

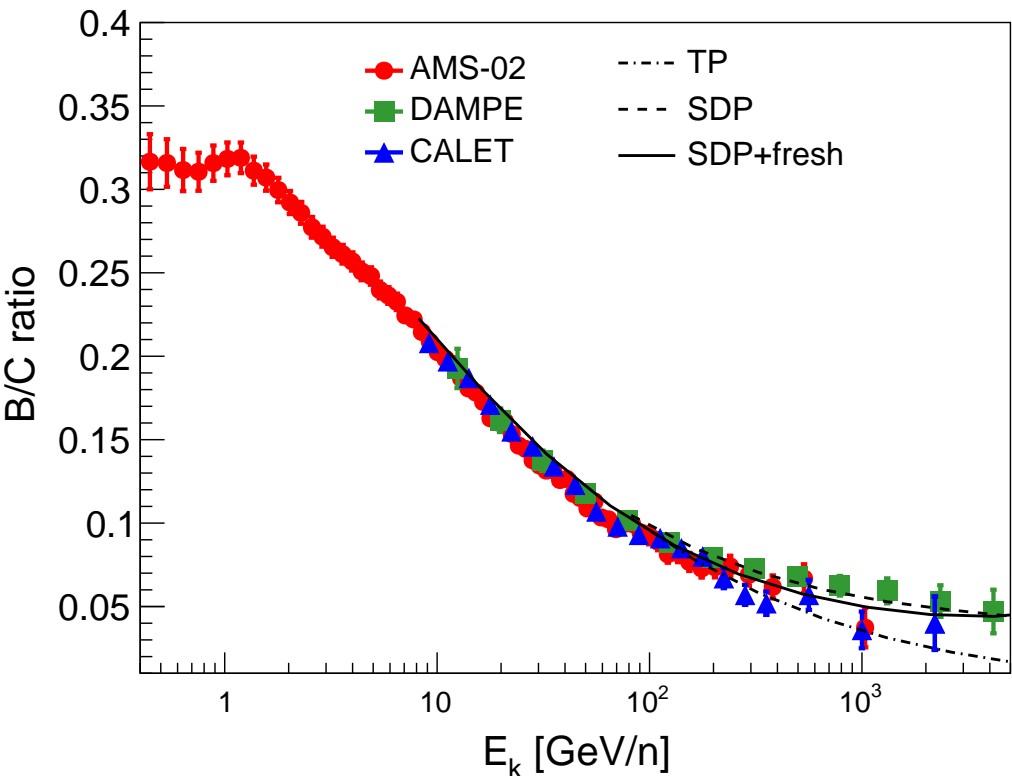

**Figure 1.** The model calculations of secondary to the primary ratio of B/C from Traditional Propagation (TP), Spatially Dependent Propagation (SDP), and SDP + "fresh" component models, compared to the measurements by AMS-02 [11], DAMPE [33] and CALET [60].

### 3.2. Helium Spectra

By means of the determinate diffusion parameters, we make an analysis for the Helium spectra of different models. The top panel of Figure 2 shows the Helium spectra expected from the three models, compared with the measurements [10]. The meaning of the different line and point styles are the same as in Figure 1. The plot starts from 60 GeV/n because AMS-02 has observed the time dependency of the Helium spectrum at lower energies due to solar modulations [61]. The model parameters for different models are given in Table 2. The TP model also cannot reproduce the hardening of the Helium spectrum above several hundred GeV. However, for the other two models, good consistency between the model and the data can be seen. It can be attributed to the effect of SDP and the summation of the background contribution and the local SNR contribution, respectively. Similar spectral

features are expected to be present for all species, as revealed recently by the DAMPE proton spectral measurement [16].

The bottom panel of Figure 2 shows the residuals between the three models and the data. The marker colors and styles correspond to the experiments, while the line styles correspond to the models. The dashed black line at 0 is to guide your eyes. As is shown in Figure 2, the SDP model gives a better result while the SDP+fresh model predicts lower helium fluxes than the DAMPE and CALET measurements above $\sim$600 GeV. However, it should be noted that the parameters of the three models are obtained by adjusting them manually to match data. It is still confirmed that the traditional propagation model is no longer applicable, while the SDP model and SDP+fresh model can be used to interpret the existing observation.

**Figure 2.** The model calculations of Helium spectrum from Traditional Propagation (TP), Spatially Dependent Propagation (SDP), and SDP + "fresh" component models. The data are from the AMS−02−publication [10], DAMPE−publication [17], and CALET−puclication [62]. $r_{model}$ and $r_{exp}$ represent the model calculated value and the measured value of the experiment, respectively.

## 4. Conclusions

CR observation have entered the era of accurate measurement, and more and more measurement results challenge the traditional CR propagation models. Previous experiments, including AMS-02, have observed a tendency for primary nuclear spectra to harden and then soften. Additionally, the B/C ratios newly measured by DAMPE also show significant hardening at similar energies, suggesting that these different spectral structures probably share a common origin.

In this work, we examine several different GCR propagation models in an attempt to explain DAMPE's new measurements of the B/C ratio and the helium spectra. As a result, we find that the traditional propagation model alone cannot reproduce the helium spectra and the B/C ratio well. Considering the recent observations of HAWC and LHAASO,

the cosmic rays are diffused non-uniformly throughout the galaxy, then the modified SDP model based on the traditional model can well reproduce the hardening of the B/C and helium nuclear spectra observed by DAMPE in the high-energy part. In addition, the primary cosmic rays can collide with the interstellar medium to produce secondary particles in the process of propagation, and also produce secondary particles with the surrounding gas when accelerated near the source region, and this contribution cannot be ignored. Considering the contribution of these two parts, the problem of excessive secondary particles is also nicely explained. Therefore, we believe that either the simple propagation effect or the propagation effect combined with the neighboring source effect can be used to explain the primary component spectrum and the secondary to primary ratio hardening of cosmic rays. We plan to extend the preliminary investigation performed in this work by conducting Bayesian analysis of fluxes for different models: (i) systematic analysis of model uncertainties, and (ii) test the model against a larger set of observables (other primary species, secondary species, etc).

**Author Contributions:** Conceptualization, Q.L. and J.F.; methodology, Q.L.; software, Q.L.; validation, Q.L. and J.F.; formal analysis, Q.L.; investigation, J.F.; resources, Q.L.; data curation, Q.L.; writing—original draft preparation, Q.L.; writing—review and editing, J.F.; visualization, J.F.; supervision, J.F.; project administration, P.-H.T.T.; funding acquisition, P.-H.T.T. All authors have read and agreed to the published version of the manuscript.

**Funding:** This work is supported by the National Natural Science Foundation of China (NSFC) through grant 11633007, the Fundamental Research Funds for the Central Universities, and the Sun Yat-sen University Science Foundation.

**Acknowledgments:** The authors thank Pengwei Zhao for the helpful discussions and for reading the manuscript.

**Conflicts of Interest:** The authors declare no conflict of interest. The funders had no role in the design of the study; in the collection, analyses, or interpretation of data; in the writing of the manuscript; or in the decision to publish the results.

## Abbreviations

The following abbreviations are used in this manuscript:

| | |
|---|---|
| AMS-02 | Alpha Magnetic Spectrometer |
| ATIC-2 | Advanced Thin Ionization Calorimeter |
| B/C | Boron-to-carbon |
| CALET | Calorimetric Electron Telescope |
| CR | Cosmic ray |
| CREAM | Cosmic Ray Energetics and Mass |
| DAMPE | Dark Matter Particle Explorer |
| ECR | Extragalactic cosmic ray |
| GCR | Galactic cosmic ray |
| HAWC | High-Altitude Water Cherenkov Observator |
| IH | Inner halo |
| ISM | Interstellar Medium |
| LHAASO | Large High Altitude Air Shower Observatory |
| OH | Outer halo |
| PAMELA | Payload for Antimatter Matter Exploration and Light-nuclei Astrophysic |
| SDP | Spatially dependent propagation |
| SNR | Supernova remnant |
| THM | Two-halo model |
| TP | Traditional propagation |

**Appendix A**

*Appendix A.1*

So far, there are two main ways to solve the cosmic ray propagation equation. The first method is to solve the propagation equation analytically (or semi-analytically) by assuming the simplified distribution of the source and interstellar gas. The second method is to use numerical solution software packages to solve the propagation equation. GALPROP [63,64] and DRAGON [65–67] are common software packages.

GALPROP is a numerical package developed by Strong, Moskalenko, and others. To better simulate the real Milky Way, GALPROP combines as much new observational data as possible with the latest theoretical developments and makes sure that the distribution of interstellar gas, radiation, and magnetic fields is as consistent as possible. In addition, GALPROP includes a variety of physical processes, such as diffusion, convection, reacceleration, energy loss, disintegration, and decay, and tries its best to restore the whole propagation process. Compared with the analytical method, it has a wide range of applications. It can simultaneously deal with the propagation of various primary cosmic rays in interstellar space and calculate the generation of secondary particles and diffuse gamma rays. Importantly, GALPROP is also versatile enough to introduce new physical effects as needed.

In previous studies, the diffusion of cosmic rays was assumed to be isotropic and uniform, i.e., applying the same diffusion coefficient throughout the propagation space. However, this assumption is not reasonable, because the diffusion coefficient is usually related to the direction of the regular magnetic field and the ratio of the energy density between the regular magnetic field and the random magnetic field. Moreover, some experimental observations have also proved that the diffusion coefficient is not uniform throughout the galaxy, which will affect the distribution of sources, the angular distribution of secondary gamma rays and neutrinos, and the anisotropy. Therefore, after considering the heterogeneity and anisotropy of diffusion coefficient in space, the developers developed a new numerical software package DRAGON (Diffusion of cosmic rays in galaxy modelization) based on GALPROP.

DRAGON adopts a second-order Cranck–Nicholson scheme with Operator Splitting and time overrelaxation to solve the diffusion equation. This provides fast a solution that is enough accurate for the average user. Some parts of DRAGON are built following GALPROP, v50p. Therefore, it kept in the code some features and models used in Galprop, such as nuclear cross-sections, gas distribution, and the convergence technique. However, each of these models is accompanied by other models, which can be selected by setting the appropriate switch.

*Appendix A.2*

This part introduces in detail how to calculate the energy spectra of the three models in our work. Before we begin, let us first make it clear that the helium nuclear spectrum of the SDP model and the traditional model contains only one component, the background component. In the SDP+fresh model, we use two different sources of cosmic rays to explain the helium nuclear spectrum (a background component and a local source component, respectively). We use different methods to calculate the energy spectra of background components and local source components.

To calculate the background energy spectrum, we use the DRAGON numerical package introduced in the previous section. We have introduced these three models in detail in the previous part of the model. The difference between the traditional model and the SDP model is the difference in the diffusion coefficient. Therefore, when we calculate the background energy spectrum of these two models, we need to modify the diffusion.cc in the DRAGON package according to their respective diffusion coefficient formulas. The SDP+fresh model uses the same program as the SDP model to calculate the background energy spectrum. The difference is that in the SDP+fresh model, we also consider the contribution of local sources.

We use an analytical algorithm to calculate the contribution of a single local source to cosmic rays. We use a time-dependent propagation equation to describe the propagation of cosmic rays injected instantaneously from a point source [68]. The injection rate as a function of time and rigidity is assumed to be

$$Q^{snr}(\mathcal{R}, t) = Q_0^{snr}(t)\left(\frac{\mathcal{R}}{\mathcal{R}_0}\right)^{-\gamma} exp\left[-\frac{\mathcal{R}}{\mathcal{R}_c^{He}}\right], \tag{A1}$$

where $\mathcal{R}_c^{He}$ is the cut-off rigidity of its accelerated CRs. It is assumed that the injection process of SNR is similar to the burst-like, and the injection rate is

$$Q_0^{snr}(t) = q_0^{snr}\delta(t - t_0), \tag{A2}$$

where the $t_0$ represents the time of the supernova explosion. Then, by convolving the time-dependent injection rate $Q_0(t)$ with Green's function, we can obtain the propagation spectrum of local SNRS

$$\phi(\vec{r}, \mathcal{R}, t) = \int_{t_i}^{t} G(\vec{r} - \vec{r}', t - t', \mathcal{R})Q_0(t')dt'. \tag{A3}$$

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
