# Peer review of "Explaining the Hardening Structures of Helium Spectrum and Boron to Carbon Ratio through Different Propagation Models"

_galaxies, doi:10.3390/galaxies11020043_

Round 1

Reviewer 1 Report

I think that the present manuscript is reasonably good in terms of quality, originality, soundness, significance, impact for being published in Galaxies. 

Author Response

We would like to thank the referee for his/her comments. We will try to improve the English in our manuscript.

Reviewer 2 Report

The work presented here is a nice study of different models for the development of the spectrum of cosmic rays and the comparison of the results of these models with actual data.

The paper is well-structured and understandable. The only thing I would like the authors to add is a detailed description of the actual programming algorithm/routines/software used to calculate the spectra. After adding this and fixing the minor issues I listed below, the paper is ready for publication.

Minor issues:

- general remark: Units (e.g. kpc, pc) shoud not be cursive

- lines 38-40: “proton spectrum have different energy indices from that of Helium, and the Helium spectrum hardens at the same energy as proton” -> proton spectrum has different energy indices than the Helium one, and that the spectrum for Helium hardens at the same energy as the one for protons”

- lines 45-46: “measurement … is also” -> “measurements … are also”

- line 51: “model” -> “models”

- line 61: “SPD model” -> “the SDP model”; “neighboring” -> “the neighboring”

- line 82: “decay.” -> “decay, respectively”

- equation (3): Please explain the meaning of eta

- line 170: “Helium spectrum” -> “Helium spectra”

- lines 171/172: “The meanings of different line styles are same as in Figure 1” -> “The meaning of the different line styles are the same as in Figure 1

- caption Fig. 2: “The data are from AMS-02 publication [34].” -> “The data are from the AMS-publication [34].

- line 181: “CR observation has” -> “CR observation have”

Reviewer 3 Report

The work aims in providing a phenomenological interpretation of the hardening featured by the B/C ratio according to the very recent DAMPE measurement, a feature which is not expected in the established, yet obsolete, models for cosmic ray origin, acceleration and propagation. B/C, being a ratio between species of CR which are mostly secondary and primary, is considered as a very sensitive observable to probe galactic propagation models and constrain their parameters. 

The scientific objective is consequently attractive, as it investigates frontier problems in cosmic ray astrophysics with possible consequences in assessing the astrophysical background for searches of beyond-standard-model physics in cosmic rays.

The introduction (Chapter 1) is enough detailed and provides a well-sounding introduction to the physics problem.

Chapter 2 describes the analytical implementation by the authors of three models, all available in literature. As also stated later, the author should consider to extend this chapter, eventually using an appendix, the provide the reader all the details to ensure the reproducibility of their results.

Chapter 3 requires an extensive rework, as it fails in describing the procedure that the authors have used to achieve their results, and does not provide any information on the robustness of their result.

Chapter 4 should also be extended and revised based on possible additional information that the authors could provide by extending the manuscript accordingly for coming suggestions.

Overall, the authors have chosen to investigate the predictive capabilities of 3 models available in the literature with increasing complexity, fixing some parameters based on other works and identifying the best-fit value of remaining relevant propagation parameters. This approach is fine, although it defines an arbitrary set of boundaries whose systematic effect in the final result is not discussed, as it should be.

The work features additional critical major flaws, as follows:

  • the study limits itself in considering only the latest B/C measurement by DAMPE and AMS-02 and tests the result only on the Helium spectrum of AMS-02. The authors do not provide a consistent explanation of this choice. This approach consequently provides only a qualitative result, because the authors do not describe its capability to fully describe a larger set of cosmic ray observables available in the literature. The authors should consider to point this out, or to extend the study to a larger set of observables, discussing the limits (if any) of the best-fit model that has been identified.
  • At the date of submission (31 december 2022), the CALET collaboration had already published the measurement of the B/C ratio (https://journals.aps.org/prl/abstract/10.1103/PhysRevLett.129.251103) that - differently from the DAMPE measurement - shows a larger level of consistency with AMS-02 data at high energies. The authors should, at least, acknowledge this measurement and test their resulting models against the CALET measurement, in order to provide the reader a consistent description of the latest measurements available in the field.
  • The description of the models implemented in this work should be improved. Several parameters are not described in the text, some appear as free parameters without being introduced to the reader.
  • Most importantly, the authors only report their best-fit parameters, and do not discuss at all the goodness of the fit, the model parameter uncertainties, and how statistical and systematic uncertainties (which could introduce correlations between data points) of the B/C and He flux data have been used in the fitting procedure. The authors should consequently largely improve the manuscript with a detailed description of the analytical procedure and  with a discussion of the robustness of the final result in terms of statistical and systematic uncertainty in the fit parameters.

As a conclusion. I suggest the author to extend the reach of their work by providing a thorough analysis of the model results, both in terms of robustness and systematic uncertainty assessment as well as interpretation of the results. Moreover, as it is, the work seems to only aim in supporting the fact that the recent DAMPE result provides additional information to assess cosmic ray origin, acceleration and propagation models that- in my opinion - should not be the main achievement of a scientific article in astroparticle phenomenology. In view of this, the authors could consider to test their model against other observables, both in terms of secondary/primary ratio as well as primary or secondary spectra (which in the current work are limited to the He flux only with no clear explanation for this choice), to provide a more general overview of the descriptive capability of their model implementation in the general cosmic ray propagation scenario. Clearly, in this case, the abstract and title should be changed accordingly.

Other suggestions and typos follow:

  • general: the “Traditional Propagation” is a common term used in the field and does it indicate unambiguously one particular model? If not, please use italic or other form to highlight the fact that this is a jargon adopted by the authors.
  • L6 - abnormal -> anomalous
  • L6 - L8: the sentences should be rewritten in a “positive” statement, e.g.: “We have verified that an extension of traditional propagation model taking into account spatially dependent propagation and secondary particle acceleration provide a more accurate description of latest DAMPE B/C data and of AMS-02 He flux data”.
  • L9: this fact has not been shown in the article, there is not discussion on parameter constraints because only best-fit value are provided.  
  • L14: a reference is needed
  • L17: … and so on. Concurrently, secondary …
  • L26: “It’s known to all” may not be true for unexperienced readers. Also, the english does not sound. I suggest to remove this incipit.
  • L27: remove “exact”
  • L31: why “nucleus”? Is the sentence about protons or about CR nuclei?
  • L32: AMS-02 has for the first time provided a detailed analysis of the rigidity dependence of the spectral features in CR nuclei spectra.  This should be acknowledged.
  • L34: CALET has not only confirmed the results, but has for the first time extended the measurement of protons in CRs from few tens of GeV to 10 TeV consistently with one instrument. This should be acknowledged.
  • L55: please extend the sentence
  • L60: please indicate that the models will be described in detail later in the text.
  • L67: please remove the first sentence, or modify the word “complicated” which is ill-posed.
  • L75: please insert a reference
  • L85: please provide a reference for the “straight-ahead” approximation.
  • -eq 4: va -> Va ?
  • L106: kpc is italic. Please consider to uniform the style of units throughout the whole document
  • eq 6: the function seems to feature a discontinuity in R=R_{br1}. This should not happen.
  • L111: why the index “1” in R_{br1}?
  • L112: in which step of the phenomenological description is the force field approximation used? The B/C should be quite independent, and the He flux is used only above 60 GV. Also, the resulting force-field parameter should be explicitely written for consistency, being part of the model.
  • eq 7: many parameters have not been introduced, e.g.: R’_{0}, \delta (which now has a r and z dependence), n, N_m. Please revise the whole manuscript taking care that all non- trivial parameters are described for the reader. You could consider to write an appendix for a detailed description of the models and related parameters employed in the work
  • L138 images -> scenarios
  • L154: please consider to elaborate on “fitting the experimental data manually”
  • L156: nulear
  • table 1: delta_0 non discussed
  • fig 2: the last AMS-02 data point seems missing the upper uncertainty bar
  • still on fig 2: an ancillary graph plotting the residuals between the model and the data could be helpful to provide a first clear feedback on the goodness of the models.
  • L184: B/C
  • L185: what is “they”, and with what they share a common origin? Please elaborate more in details the conclusions.
  • L186: the models are not based on DAMPE’s new measurement of B/C. They are tested against it.
  • L187: you have only tested the B/C ratio, not the sparate B and C spectra. Please revise the sentence accordingly
  • L190: “so the modified”… this sentence is not clear. L190 is not a consequence of L1189, and the word “so” implies. Please revise the sentence
  • L196: the word “perfectly” is not adequate unless a more robust analysis and detailed understanding fo the results is presented to the reader.

Round 2

Reviewer 3 Report

The authors have revised the manuscript according to the suggestions provided during the first round of review, and the quality of the manuscript is largely improved.

I acknowledge that the authors are planning to perform a systematic study of model uncertainties and extend the set of observables used to test the models in a future publication, as strongly suggested during revision n.1. This approach would have strongly increased the scientific relevance of this work, and hopefully will be the objective of a prompt follow-up publication.

The status of the manuscript and its content is currently potentially acceptable for publication after minor revisions, which are indicated as follows.

I also strongly urge the authors to check the spelling of english words and typos, which is poor in many places, especially in newly added sentences

Answer 4: please indicate this in the document, because it justifies the absence of an uncertainty analysis in the document.

Answer 25: Does the use of the force filed approximation model, which applies at low energies, modifies the result of the B/C fit and of the He fit? If so, then it is ok to leave it here, otherwise the description could be removed.

abstract, L8: “more accurate description of latest DAMPE B/C data 8 and of AMS-02 He flux data” -> “more accurate description of latest B/C and of He flux data by DAMPE, CALET and AMS-02”

abstract, L9: “which can help to set certain tighter constraints on the propagation 9 parameters in the future work“. The meaning of this sentence is not clear

L60: “These observations ma indicate that CRs have a common origin, the propagation effect or the source effect.”. The sentence is not clear

L104: “In order to better fit the measurements, we introduce a correction 104 factor η for the speed of light“. The approach is not clear. The author should instead describe the physical meaning the complex \beta^\eta dependence in the diffusion coefficient. This short explanation is impossible to interpret. 

L169: “so we chose it in our work” -> £so has been chosen to best constrain the parameter models tested in this work”

L170-172: time constraints cannot be accepted as an explanation for a limited analysis. You could rewrite the sentence in a positive manner, such as (please modify accordingly). “Although this work is presently limited to the analysis of ….., the extension of the analysis to other nuclei fluxes such as …. will be the objective of a future publication”.

L175-L178. The new sentence that the authors have added should be rewritten more clearly, because it does not give a proper indications of the approach that has been used. Every reader should, in theory, be able to reproduce the result, and the authors should provide the relevant amount of information to allow this. Is the technique used well known in literature? If so, please indicate a reference. What’s the meaning of “again and again”? How did the authors define the convergence of their fit procedure? Finally, the word “manually” is very confusing, please use another word to indicate the procedure.

L 183: why “relevant”?

L197: “DAMPE, CALET and AMS-02 data”. Please acknowledge CALET results together with DAMPE when the same data are used or fitted together.

L218-220: please use other wording for “low”, which is ill used in this case.

L213-224: This paragraph provides one of the strongest conclusions of the work, and I suggest the authors to work on the wording to better represent their achievements. 

Figure 2: please indicate the meaning of “r” in the caption.

L228: please use AMS-02 throughout the text.

Conclusion paragraph: the conclusion paragraph must clearly contain a sentence that expresses clearly the follow-up works declared by the authors that are planned to extend the preliminary investigation performed in this work: i) systematic analysis of model uncertainties ii) test the model against a larger set of observables (other primary species, secondary species, ….)

Appendix  1: please cite GALPROP and DRAGON. 

L299-304: I suggest the authors to remove this sentence as it does not provide additional information to the results of this work
